# A Familial Case of Multiple Endocrine Neoplasia 2A: From Morphology to Genetic Alterations Penetration in Three Generations of a Family

**DOI:** 10.3390/diagnostics13050955

**Published:** 2023-03-02

**Authors:** Lan Chen, Jing-Xin Zhang, Dong-Ge Liu, Hong-Gang Liu

**Affiliations:** 1Pathology Department, Beijing Hospital, National Center of Gerontology, Institute of Geriatric Medicine, Chinese Academy of Medical Sciences, Beijing 100730, China; 2Pathology Department, Beijing Tongren Hospital, Capital Medical University, Beijing Key Laboratory of Head and Neck Molecular Diagnostic Pathology, Beijing 100730, China

**Keywords:** medullary thyroid carcinoma, pheochromocytoma, multiple endocrine neoplasia type 2A, RET germline mutation

## Abstract

This paper illustrates a rare syndrome of multiple endocrine neoplasia type 2A (MEN2A) in a family of three generations. In our case, the father, son and one daughter developed phaeochromocytoma (PHEO) and medullary thyroid carcinoma (MTC) over a period of 35 years. Because of the metachronous onset of the disease and lack of digital medical records in the past, the syndrome was not found until a recent fine needle aspiration of an MTC-metastasized lymph node from the son. All resected tumors from the family members were then reviewed and supplemented with immunohistochemical studies, previously wrong diagnoses were then corrected. Further molecular study of targeted sequencing also revealed a RET germline mutation (C634G) in the family tree including the three members with onset of the disease and one granddaughter who had no disease at the time of testing. Despite the syndrome being well-known, it may still be misdiagnosed because of its rarity and long disease onset. A few lessons can be learned from this unique case. Successful diagnosis requires high suspicion and surveillance and a tri-level methodology including a careful review of family history, pathology and genetic counselling.

**Figure 1 diagnostics-13-00955-f001:**
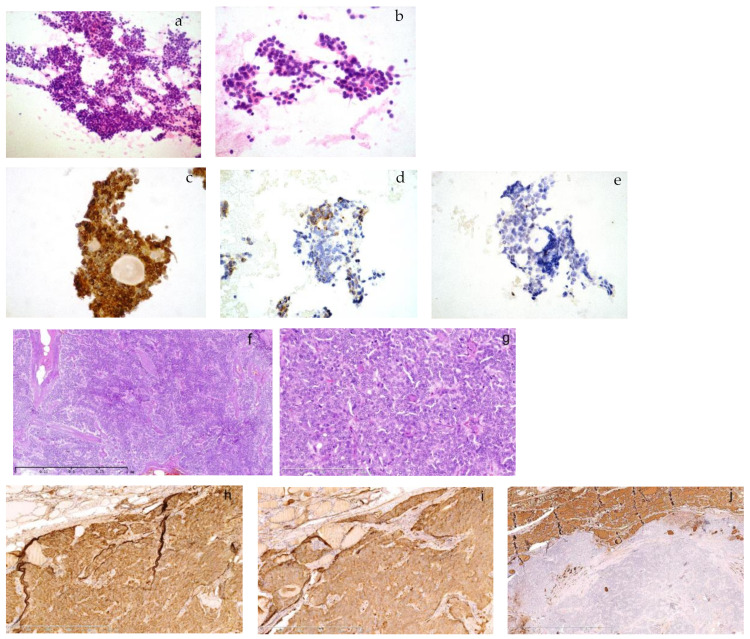
The cytology of a metastasized medullary thyroid carcinoma (MTC) in the neck lymph node and its primary MTC in the son born in 1968. A 49-year-old male in the outpatient clinic had a right lymphadenopathy, 1.2 cm in diameter, in area III. He had ultrasound-guided fine needle aspiration. The smear was composed of a large amount of monomorphic, small, round cells in large aggregates or dispersed ((**a**), HE, 100×). The cells were plasmacytoid with eccentric nuclei, which had fine chromatin or small nucleoli. There was no mitosis or necrosis ((**b**), HE, 400×). Immunohistochemistry on smears showed strong immunopositivity for chromogranin A ((**c**), 400×), TTF-1 and CEA (not shown). There was moderate positivity for calcitonin ((**d**), 400×) and no expression of thyroglobulin ((**e**), 400×). It was consistent with metastasized thyroid medullary carcinoma. He was diagnosed with bilateral multiple MTC with right neck lymph node metastasis in 1/2 in our hospital 8 years ago when he was 41. Tumors infiltrated thyroid tissues in a solid and trabecular growth pattern and there was amyloid deposition in the background ((**f**), 50×). It was composed of small monomorphic cells with fine chromatin or small nucleoli in a syncytial pattern ((**g**), 200×). Immunohistochemistry showed a strong expression of chromogranin A ((**h**), 100×) and calcitonin ((**i**), 100×) and absence of thyroglobulin ((**j**), 25×). The cell morphology of his recent lymph node aspiration was consistent with his previous pathology. The patient was then admitted into the ward for further treatment. A family syndrome of multiple endocrine neoplasia type 2A (MEN2A) surfaced when we dug into his family history and previous pathology. His father and one of his elder sisters used to have intermittent hypertension, cardiac palpitation, dizziness and epileptic seizure, etc. He was 31 when he first suffered similar symptoms and was then hospitalized and diagnosed with left adrenal pheochromocytoma (PHEO) and right thyroid adenoma. We suspected the thyroid adenoma might have been misdiagnosed as application of immunohistochemistry was in its early stage in the late 1990s during his first onset of the disease. His pathology materials were not kept due to the long time period in a different local hospital where he received surgical resections. We proceeded to review the pathology of his father and sister who had surgeries in our hospital.

**Figure 2 diagnostics-13-00955-f002:**
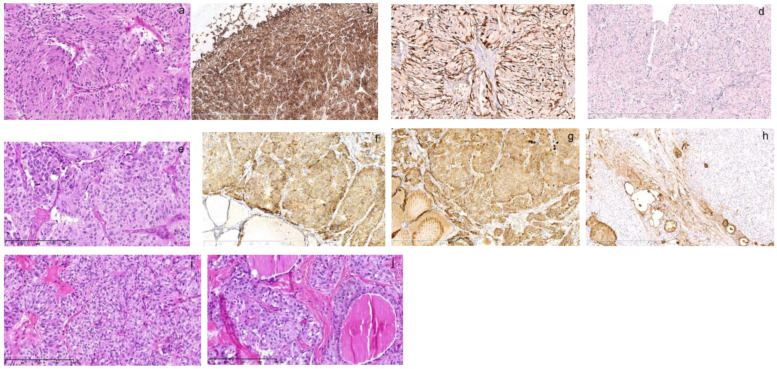
The pathology of thyroid and adrenal gland tumor of his elder sister born in 1964. One of his elder sisters, who was 4 years older than him, was diagnosed with left adrenal PHEO and right thyroid follicular carcinoma at the age of 28 and left thyroid MTC at 45. As per her previous pathology, her PHEO diagnosed at 28 ((**a**), HE, 200×) was composed of zellballen-like nests of tumor cells separated by vascular sinuses. The tumor cells were spindle-like with amphophilic granular cytoplasm. The small and monomorphic nuclei were vacuolar with small nucleoli. Immunohistochemistry showed that tumor cells were positive for chromogranin A ((**b**), 50×). The small spindle-like sustentacular cells were outlined by S100 positivity at the periphery of tumor nests ((**c**), 200×). The proliferative index was less than 1% as indicated by Ki67 ((**d**), 100×). Her left MTC diagnosed at 45 ((**e**), HE, 200×) was composed of trabecular solid growth patterns of spindle or polygonal tumor cells. The tumor cells had round nuclei that varied in size with amyloid substance deposition in the background. They were strongly positive for chromogranin A ((**f**), 100×) and calcitonin ((**g**), 100×) and negative for thyroglobulin ((**h**), 100×). When we reviewed her right thyroid follicular carcinoma diagnosed at 28 ((**i**), 200×), it was also composed of a similar pathology as her left MTC showing nests of syncytial cell aggregates with monomorphic, round nuclei. There was also nuclei variation in size ((**j**), 200×). Supplementary immunohistochemistry demonstrated immunopositivity of chromogranin A and calcitonin (not shown). The thyroid follicular carcinoma diagnosis was wrong and it should be MTC instead. Therefore, the elder sister had both MTC and PHEO in a time span of 17 years. The misdiagnosis was not recognized earlier as her first medical record in early 1990s was not digitalized and was disconnected with her second record until we manually discovered it this time. The syndrome was missed then due to the metachronous onset of the disease that was misdiagnosed as thyroid adenoma or follicular carcinoma in the brother and sister. MTC can mimic any thyroid tumor and requires immunohistochemistry for accurate diagnosis; however, immunohistochemistry back then was not comprehensive.

**Figure 3 diagnostics-13-00955-f003:**
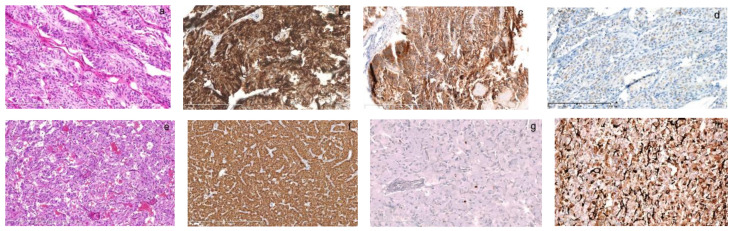
The pathology of left MTC and left adrenal PHEO in the father diagnosed in our hospital. Born in 1940, the father had surgeries for PHEO and MTC at 42. His previous medical file and pathology were searched manually, reviewed carefully and supplemented with immunohistochemistry. The left thyroid tumor was 4 × 3.5 × 2.5 cm in size and composed of a trabecular growth pattern of spindle-like cells with monomorphic, round or spindle nuclei ((**a**), HE, 200×). They were strongly positive for calcitonin ((**b**), 200×) and CEA ((**c**), 200×), and weakly positive for chromogranin A ((**d**), 200×) and the diagnosis was consistent with MTC. His PHEO was composed of nested polygonal cells with abundant amphophilic cytoplasm separated by rich vascular sinuses ((**e**), HE, 100×). They were strongly positive for chromogranin A ((**f**), 100×) and sparsely positive for Ki67 ((**g**), 200×). The sustentacular cells at the periphery of tumor cells were strongly positive for S100 ((**h**), 200×). The syndrome was missed because our knowledge for this rare syndrome was limited in early 1980s. His children had not had onset of the disease back then.

**Figure 4 diagnostics-13-00955-f004:**
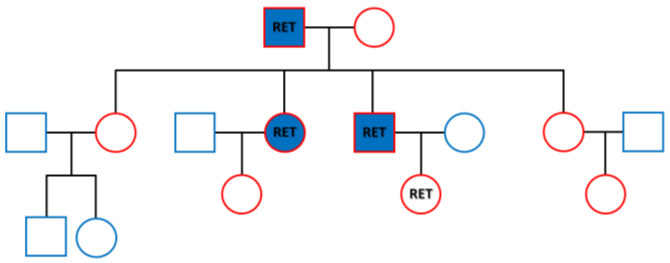
RET germline mutation detected in the family of three generations. In 1968, Steiner first reported multiple endocrine neoplasia type 2. The syndrome is classified into MEN2A, 2B and familial MTC depending on the presence and type of additional symptoms. Virtually all MEN2A patients have MTC and 50% of them have concurrence of PHEO, and 15–30% have hyperparathyroidism. It accounts the most common variation of MEN2 with an incidence of 1/1,973,500 [1]. The major molecular alterations of RET in MEN2A lie in codons 609, 611, 618 or 620 of exon 10 or codon 634 of exon 11. The latter accounts for 80% of germline mutations of MEN2A [2]. For our cases, 4 mL blood was taken for each person from the three generations of the family as shown in the family tree. It included the parents, their four children and two grandchildren outlined in red. DNA was purified using a QIAmp Blood DNA kit (Qiagen, Germantown, MD, USA). Targeted next-generation sequencing was successfully performed for the RET proto-oncogene by Gene Plus, Beijing, China, according to manufacturer’s protocol. A missense mutation c.T1900G in exon 11 causing protein alteration of p.C634G was discovered in all three members with onset of the disease and one granddaughter who had no disease at the time of testing. The family members with the mutation are labeled as RET in the figure, while those with onset of MEN2A are labeled in blue. The granddaughter with the RET mutation, was 18 years old and is the daughter of the son. A close monitoring was undertaken for the girl according to the American Thyroid Association guidelines for adults, that requires surveillance of serum calcitonin [1]. The family tragedy might be stopped thereafter. MEN2-associated RET mutations have a gain of function effect to promote activation of the kinase and oncogenic conversion via downstream signaling pathways [3]. RET tyrosine kinase inhibition will provide targeted therapy for the family members with established RET-positive cancers [4]. Selpercatinib, one kinase inhibitor, has been approved by the U.S. Food and Drug Administration for patients with advanced or metastatic RET-mutant MTC [5]. It inhibits wild-type RET and multiple mutated RET isoforms and may benefit the son who had repeated recurrence of the cancer. In conclusion, misdiagnosis of thyroid tumor and metachronous onset of the disease in the family hampered our recognition of this well-documented but rare disease. A few lessons can be learned from this case. High suspicion was a critical starting point in our diagnosis. A careful review of previous medical records helped connect the dots. Molecular screening then played a key role in reaching our eventual diagnosis.

## Data Availability

The authors declare that all datasets on which the conclusions of the paper rely are available to editors and reviewers without unnecessary restriction. The datasets can be obtained from the corresponding author Lan Chen on reasonable request.

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
