# Peer review of "A Familial Case of Multiple Endocrine Neoplasia 2A: From Morphology to Genetic Alterations Penetration in Three Generations of a Family"

_diagnostics, 2023, doi:10.3390/diagnostics13050955_

Round 1

Reviewer 1 Report

Thank you for the opportunity to review this manuscript. The author have presented an interesting report of undiagnosed MEN2A syndrome in family. This is a syndrome that is well described and well taught even at a medical student level. Yet, it can remain undiagnosed affected quality of life and mortality in patients across families. This delay in diagnosis of this well known entity is worth re-iterating to emphasize the need for high suspicion and surveillance. However, the text of this manuscript is need of considerable editing. The content of the manuscript should be structured with headings for a better read. The overall flow of the case needs better narration. This manuscript will also benefit from a brief review of presentation, surveillance and treat modalities for this syndrome. I would strongly recommend adding a discussion section highlighting the lessons learned from this case and take home points for the readers at different stages of learning. 

Author Response

Thank you for your objective and pertinent comments. As you acutely observed, although it is a well-documented rare syndrome, it still took us 35 years to diagnose the disease because of previous misdiagnosis and incomplete medical histories during the long metachronous onset of the disease amongst the family. A few lessons can be learned from this case. High suspicion was indeed a critical starting point in our diagnosis. A careful review of previous medical records helped connect the dots. Molecular screening then played a key role in reaching our eventual diagnosis. We chose the format of Interesting Images to share this unique case, which are visually illuminating and educational for young pathologists. As you suggested, we restructured the flow of narration and added subtitles for each figure panel to highlight the key information. A brief introduction of MEN2A was added throughout lines 166-171. A separate discussion is not allowed by the format, so we included it in the last figure panel (lines 190-194).

Reviewer 2 Report

Dear Authors, you present an interesting familial MEN2A cases. I have some comments on it. 

Structural problem: The main text is under "Figure" headings. Figure explanations are required for the histological pictures, but the main text can tell the family story. Please re-edit the texture.

Minor changes: Page 5 Line 158-160. RET-inhibitors are not suitable for "prevention" strategies

Author Response

Thank you for your appreciation and comments. The format of interesting images is restrictive and can only include figures and legends. So, we told the family story in the abstract and through the legends. The flow of narration has now been restructured to emphasize pathology and molecular alteration of the disease and lessons learned from this case. The mechanisms of RET mutation in the pathogenesis of MEN 2 was briefly illustrated in lines 183-186. RET targeted therapy was also re-written in lines 186-189.

Round 2

Reviewer 1 Report

Thank you for the opportunities to re-review this manuscript. The authors have appropriately addressed my comments. 

Author Response

We thank the reviewer for his comments and are pleased to know he is satisfied with our revision.